# Tumor Accumulation of PIP-Based KRAS Inhibitor KR12 Evaluated by the Use of a Simple, Versatile Chicken Egg Tumor Model

**DOI:** 10.3390/cancers14040951

**Published:** 2022-02-14

**Authors:** Yuya Higashi, Shuji Ikeda, Kotaro Matsumoto, Shinsuke Satoh, Aoi Komatsu, Hiroshi Sugiyama, Fuyuhiko Tamanoi

**Affiliations:** 1Institute for Integrated Cell-Material Sciences, Institute for Advanced Study, Kyoto University, Kyoto 606-8501, Japan; higashi.yuya.5n@kyoto-u.ac.jp (Y.H.); matsumoto.kotaro.5r@kyoto-u.ac.jp (K.M.); komatsu.aoi.6z@kyoto-u.ac.jp (A.K.); 2Department of Chemistry, Graduate School of Science, Kyoto University, Kyoto 606-8502, Japan; s-ik@kc5.so-net.ne.jp (S.I.); shinsuke.sato@elsevier.com (S.S.); sugiyama.hiroshi.3s@kyoto-u.ac.jp (H.S.)

**Keywords:** CAM assay, pyrrole-imidazole polyamide, KRAS inhibitor, tumor accumulation, nuclear localization

## Abstract

**Simple Summary:**

One of the goals of nanoplatform-based cancer therapy is to achieve tumor accumulation of anticancer agents. We have focused on PIP-based KRAS inhibitor KR12 (pyrrole–imidazole polyamide indole-seco–CBI conjugate), which has been reported to exhibit tumor growth inhibition in a xenograft mouse model. To evaluate the tumor accumulation property of KR12, we have synthesized a fluorescently labeled KR12 derivative (KR12-TAMRA) and employed a chicken egg tumor assay, a simple and versatile assay to examine tumor accumulation. Our results show that KR12-TAMRA accumulates specifically in the tumor when injected into a fertilized chicken egg transplanted with human cancer cells. We also demonstrate nuclear accumulation of KR12-TAMRA. Finally, inhibition of tumor growth in the chorioallantoic membrane (CAM) assay is shown. These results uncover a number of attractive features of PIP-based KR12 for cancer therapy.

**Abstract:**

Background: The KRAS inhibitor KR12, based on pyrrole-imidazole polyamide (PIP), has been developed and shown to exhibit efficacy in mouse experiments. Because some PIP species exhibit tumor accumulation capability, we decided to evaluate whether the PIP portion of KR12 exhibits tumor accumulation. We employed the CAM assay that provides a simple method for tumor accumulation evaluation. Methods: KR12 PIP was synthesized and conjugated to TAMRA to produce a fluorescently labeled reagent (KR12-TAMRA). This reagent was injected into a fertilized chicken egg that has been transplanted with human cancer cells. Distribution of the red fluorescence was examined by cutting out tumor as well as various organs from the embryo. Results: The red fluorescence of KR12-TAMRA was found to overlap with the green fluorescence of the tumor formed with GFP-expressing cancer cells. We also observed nuclear localization of KR12-TAMRA. Treatment of KR12 that contained the alkylating agent CBI in the tumor-bearing chicken egg resulted in tumor growth inhibition. Conclusions: KR12 contains a PIP that has two key features: tumor accumulation and nuclear localization. KR12 conjugated with CBI exhibits inhibition of tumor growth in the CAM model.

## 1. Introduction

Achieving tumor accumulation of anticancer agents has been a major challenge in cancer therapy [1]. A variety of approaches have been taken to come up with an optimum strategy. For example, antibodies that recognize cancer cell surface receptors have been used [2]. Another approach is to use nanoparticles that could achieve tumor accumulation by a passive EPR (enhanced permeability and retention) mechanism [3]. The nanoparticles can also be targeted to the tumor by equipping with ligands that can target cancer receptors [4]. We have been interested in pyrrole–imidazole polyamide (PIP) compounds that can bind to minor grooves of DNA and shut down gene expression [5,6]. Interestingly, tumor accumulation of some PIP compounds has been reported [7,8].

PIPs are originally derived from the natural antibiotics netropsin and distamycin A [9,10]. They are cell permeable, can be localized to the cell nucleus, and can bind to a minor groove of DNA [5,11]. This ability was exploited to shut down gene expression for cancer therapy [5,12,13]. Of particular interest is KR12 (pyrrole–imidazole polyamide indole-seco–CBI conjugate), an inhibitor of *KRAS* gene expression [12]. This compound was designed to bind preferentially to the *KRAS* gene with a codon-12 mutation. Upon binding, CBI (1,2,9,9a-tetrahydrocyclopropa[1,2-c]benz[1,2-e]indol-4-one) alkylates DNA and blocks gene expression. Because some PIP species were reported to exhibit excellent tumor accumulation in a mouse model [7,8], we decided to investigate whether the PIP portion of KR12 has tumor accumulation property.

In this work, we employed a chicken egg tumor model (CAM model) to examine the tumor accumulation capability of KR12 PIP. The CAM model has emerged as a simple and versatile model to evaluate the tumor accumulation property of nanomaterials [14,15,16,17,18,19,20,21,22,23,24,25,26]. The CAM model uses fertilized chicken eggs. Due to the nutrient-rich and highly angiogenic nature of the chorioallantoic membrane (CAM) of a fertilized chicken egg, transplantation of human cancer cells results in the rapid formation of histochemically identifiable tumors within several days. The tumor formation is possible, partly due to incomplete immune system established at this point of chicken development. In addition, the rich vascular and nutrient state of the CAM membrane enables rapid formation of the tumor. We have used this assay to demonstrate tumor accumulation of biodegradable mesoporous silica nanoparticles [14,19]. In these experiments, we observed excellent tumor accumulation of fluorescently labeled nanoparticles after intravenous injection while doxorubicin was distributed in every organ of the chick embryo.

In this work, we used the PIP portion of KR12 and conjugated to fluorescent TAMRA to produce KR12-TAMRA and demonstrated that KR12-TAMRA strongly accumulates in the tumor in the CAM model. The accumulation is highest 1 or 2 days after intravenous injection. KR12-TAMRA also shows excellent nuclear localization property. The tumor growth inhibition property of KR12 that contains a DNA alkylating agent was demonstrated using the CAM assay. These results point to remarkable features of the PIP-based KRAS inhibitor.

## 2. Materials and Methods

### 2.1. Cells and Reagents

OVCAR8 (human ovarian cancer cells) expressing GFP (OVCAR8-GFP) were grown in RPMI1640 medium supplemented with 10% FBS (Fetal Bovine Serum, Thermo Fisher Scientific, Waltham, MA, USA) and 1% penicillin/streptomycin (Nacalai Tesque, Kyoto, Japan). Lung cancer cell line A549 cells were grown in DMEM (Dulbecco’s Modified Eagle Medium, Nacalai Tesque, Kyoto, Japan) supplemented with 10% FBS and 1% penicillin/streptomycin.

### 2.2. Synthesis of KR12-TAMRA

Reversed-phase (RP) HPLC purification was performed on an Engineering PU-2089 plus series system (Jasco, Tokyo, Japan) using a COSMOSIL 5C18-MS-II column (Nacalai Tesque, Kyoto, Japan). 0.1% TFA (Nacalai Tesque, Kyoto, Japan) in water and acetonitrile were used as the eluent with detection at 254 nm. Matrix assisted laser desorption/ionization time-of-flight mass spectrometry (MALDI-TOF MS) analysis was conducted on a microflex system (Bruker Daltonics K.K., Kanagawa, Japan).

The PIP portion of KR12 (KR12-PIP) was synthesized as reported previously [12]. Briefly, solid phase synthesis was performed on Py-oxime resin by PSSM-8 peptide synthesizer (Shimadzu, Kyoto, Japan). The products were then cleaved from the resin with N,N-dimethylpropanediamine (55 °C, 3 h). The crude sample was purified by RP HPLC. The purified KR12-PIP and 5(6)-TAMRA-X, SE (Invitrogen, Waltham, MA, USA) were conjugated to obtain KR12-TAMRA. To a mixture of KR12-PIP (9.7 mg, 5.2 μmol), 5(6)-TAMRA-X, SE (5.0 mg, 7.8 μmol), and DMF (200 μL) (Nacalai Tesque, Kyoto, Japan) was added Et3N (8.6 μL, 62 μmol) (Nacalai Tesque, Kyoto, Japan), and the reaction mixture was shaken at room temperature for 2 h. RP HPLC purification and lyophilization give KR12-TAMRA (4.6 mg, 1.9 μmol, 37%) as a purple solid. MALDI-TOF MS m/z calculated for C116H135N38O21 + [M + H] + 2396.1, found 2397.5. HPLC analysis and the MALDI TOF mass spectrum are shown in Appendix A.

### 2.3. Establishment of CAM Model

Fertilized white chicken eggs (purchased from Goto Hatchery, Gifu, Japan, or Japan Layer K.K., Gifu, Japan) were incubated in a rotary incubator at 37.5 °C and 65% humidity. After a 10-day incubation, they were used to transplant the cancer cells. A window was opened on the eggshell and the CAM was dropped. To transplant the cancer cells, we used a sterile Teflon ring that was placed at a Y-shape blood vessel on the CAM. Cells (2 × 10^6^ cells) were added into the ring and then the window was covered with Tegaderm film (3M Japan, Tokyo, Japan). An Olympus SZX12 stereomicroscope (Olympus, Tokyo, Japan) was used to observe tumor formation. After cutting out the CAM tumors, they were fixed with 4% paraformaldehyde overnight. All chicken egg experiments were approved by the Kyoto University Animal Research Committee and were performed in compliance with the committee guidelines. In vivo experiments do not require any special additional allowance as long as the embryos are sacrificed before hatching, as was done in this study.

### 2.4. Tumor Accumulation of KR12-TAMRA in the CAM Model

KR12-TAMRA (1, 5 or 25 μg) was injected intravenously into the chicken egg, and then tissue distribution of KR12-TAMRA was investigated by red fluorescence of KR12-TAMRA using a fluorescent stereomicroscope. RITC (Rhodamine-B isothiocyanate, 25 μg) was used as a control. In addition, the CAM tumor was fixed overnight with 4% paraformaldehyde at 4 °C, and its tumor (after washing with ice-cold PBS) was treated with 99.8% methanol for 30 min at −80 °C and washed in ice-cold PBS again. The tumor was incubated in a 20% sucrose solution overnight at 4 °C. Thin sections (sliced with 30 µm in thickness by the cryomicrotome) were prepared from the tumor and observed by using a confocal laser microscope (Nikon First-Scan Confocal Microscope A1R) which was equipped with a 10× lens (CFI Plan Apo 10, Nikon, Tokyo, Japan). The wavelength of excitation (Exc) and fluorescence emission (Emis) was GFP, Exc at 488 nm and Emis at 500–550 nm; TAMRA, Exc at 561 nm and Emis at 570–620 nm.

### 2.5. Nuclear Localization of KR12-TAMRA

Lung cancer cell line A549 cells (5 × 10^3^ cells) were inoculated and overnight cultured with DMEM culture medium for 24 h at 37 °C humidified CO_2_. KR12-TAMRA (final 1 nM in the plate) was added to culture medium to be taken in the cells and cultured for 6 h, again. Cells were washed with ice-cold PBS and overnight fixed with 4% paraformaldehyde at 4 °C. Nuclei were stained with Hoechst 33258 dye for 30 min in dark, and nuclear localization of KR12-TAMRA was observed by confocal laser microscope.

### 2.6. KR12 Treatment of the CAM Tumor

Fertilized chicken eggs were transplanted with human lung cancer A549 cells to form tumor on CAM. KR12 (5 μg) was intravenously injected into chicken eggs and the effect on tumor growth was investigated. Tumor was cut out and tumor weight was measured. The data are presented as the mean ± standard error of the mean (SEM) of three replicates.

### 2.7. Statistical Analysis

For the time-course experiment of CAM tumor accumulation of KR12-TAMRA, 6 and 3 samples were analyzed for Day 1 and Day 2 after injection, respectively. For CAM tumor accumulation assay of KR12-TAMRA with different concentrations, 5, 7, and 5 samples were analyzed for 1, 5, and 25 μg of KR12-TAMRA injections, respectively. For the CAM tumor growth inhibition assay using KR12, three samples were analyzed for Day 3 and Day 4 after injection, respectively. All results were represented as the mean ± SEM for each sample. The *p*-value was calculated by a two-tailed Student’s t-test using R software.

## 3. Results

### 3.1. Synthesis of KR12-TAMRA

KR12 (pyrrole-imidazole polyamide indole-seco-CBI conjugate) is a PIP-based inhibitor of KRAS expression designed to inhibit expression of the mutant *KRAS* gene [12]. KR12 binds to a DNA minor groove by the action of the PIP and alkylates DNA by the action of CBI that catalyzes the alkylation of N3 adenine. This results in the inhibition of the RAS signaling and inhibition of tumor growth in mouse xenograft models [12]. Because some PIP species can accumulate in tumors [7,8], we investigated whether the PIP portion of KR12 has a tumor accumulation property. To investigate this tumor accumulation property, we used the PIP contained in KR12 and conjugated TAMRA that conferred a fluorescent property, thus producing KR12-TAMRA. Structures of KR12 and KR12-TAMRA are shown in Figure 1. To produce KR12-TAMRA, the alkylating group was removed from one end of KR12 and TAMRA was added to the other end of KR12. Synthesis of KR12-TAMRA is described in the Materials and Methods and the results of the characterization by HPLC and mass spectroscopy are described in Appendix A.

### 3.2. KR12-TAMRA Exhibits Excellent Tumor Accumulation in the CAM Assay

To examine the tumor accumulation property of KR12, we decided to employ the chicken egg tumor model called the CAM assay. In this assay, fertilized chicken eggs are incubated at 37.5 °C and at 65% humidity to develop an embryo. By Day 10, the inside of the chicken egg looks like that shown in Figure 2A, and the embryo is covered by the nutrient-rich membrane called a chorioallantoic membrane (CAM). A window is made on the egg shell and cancer cells are transplanted on the CAM and a tumor is formed in several days. In our experiment, we used the GFP-expressing ovarian cancer cells OVCAR8 so that the tumor formed will have green fluorescence. KR12-TAMRA, which has red fluorescence, was injected into one of the veins that populate the CAM membrane.

Figure 2B shows the result one day after the injection. The photo is taken from the top of the egg through the window that was made for transplantation. The overlap of green and red fluorescence can be seen. A similar result was observed two days after the injection.

### 3.3. Preferential Tumor Accumulation of KR12-TAMRA

To further investigate the tumor accumulation of KR12-TAMRA, we cut out the tumor as well as various organs from the embryo at one day after injection and examined distribution of red fluorescence. As shown in Figure 3A, the red fluorescence of KR12-TAMRA overlaps with the green fluorescence of the tumor formed by transplanting the GFP-expressing cancer cells. In contrast, various organs from the embryo, including the heart, liver, spleen, lung, brain, intestine, and stomach, showed little red fluorescence. A low-level red fluorescence was detected in the kidney. Thus, preferential tumor accumulation of KR12-TAMRA was observed. This is quite remarkable, as most chemicals injected in a similar manner distributes to all organs, as shown in the control. In this case, we injected RITC and examined its distribution by following its red fluorescence, and we observed red fluorescence in all organs (Figure 3B).

To further investigate the tissue distribution of KR12-TAMRA, we then cut out the tumor, made thin sections, and then examined fluorescence by using confocal microscopy. As shown in Figure 4, we observed red fluorescence in the tumor that overlaps with the green fluorescence of the GFP-expressing cancer cells.

Tumor accumulation of KR12-TAMRA was further investigated by quantitating the amount of fluorescence in the tumor as well as in other organs. Figure 5A shows the time course of tumor accumulation. As can be seen, preferential accumulation in the tumor was observed 1 day after the injection and the accumulation increased by two days after the injection. A slight fluorescence detected in kidney was near the detection limit. Figure 5B shows tissue distribution when a different amount of KR12-TAMRA was injected. Preferential tumor accumulation was observed when 1 μg or 5 μg of KR12-TAMRA was injected. However, when 25 μg was injected, we observed distribution in other organs such as the kidney, intestine, or liver.

### 3.4. KR12-TAMRA Exhibits Nuclear Localization Property

By using KR12-TAMRA, we uncovered another remarkable property of KR12 PIP. This concerns the nuclear localization of KR12. Pyrrole–imidazole polyamide (PIP) compounds can accomplish nuclear localization upon entering cells. The efficacy of cellular uptake and nuclear localization appears to depend on the structure of PIP. To investigate the nuclear localization of KR12, we added KR12-TAMRA to the culture media of lung cancer cells and incubated them for one day. Figure 6A shows efficient uptake of KR12-TAMRA into these cells shown at low magnification. Figure 6B shows higher magnification of cells. Excellent overlap of the red fluorescence of KR12-TAMRA is seen with blue fluorescence of Hoechst nuclear staining. Thus, KR12 shows an excellent nuclear localization property.

### 3.5. Inhibition of Tumor Growth in the CAM Model by KR12

KR12 contains a CBI moiety that promotes alkylation of DNA. This leads to the inhibition of cancer cell growth as well as inhibition of tumor growth in a xenograft model [12]. We used the CAM model to evaluate the tumor growth inhibition property of KR12. For the CAM tumor, we used the tumor formed by transplanting lung cancer cell line A549, as these cells carry the mutant *KRAS* gene [27]. After the tumor is formed, we intravenously injected KR12 and followed the effect on the tumor. Figure 7 shows the result of tumor growth at 3 and 4 days after the injection. Tumor growth was inhibited when KR12 was injected, as observed by the size and weight of the tumor. In contrast, the tumor continued to grow in the control where no KR12 was injected.

## 4. Discussion

In this paper, we report the excellent tumor accumulation property of the PIP-based agent KR12-TAMRA. This chemical was initially designed as an anticancer drug that binds to the mutant *KRAS* gene and was shown to exhibit tumor growth inhibition in mouse xenograft studies [12]. We used the CAM assay, a simple versatile assay that uses fertilized eggs, and uncovered the excellent tumor accumulation capability of the PIP contained in this chemical. We took advantage of the green fluorescence of the CAM tumor formed by using cancer cells expressing GFP. On the other hand, we labeled PIP with TAMRA that provided red fluorescence. Excellent overlap of green and red fluorescence was detected in the CAM assay by fluorescence microscopy. We then cut out the tumor as well as various organs from the embryo, made thin sections, and examined these by fluorescence microscopy. The results showed preferential tumor accumulation of KR12-TAMRA with little red fluorescence in other organs except the kidney. When tissue distribution was examined in relation to time course and the amount of KR12-TAMRA injected, we found that the tumor accumulation persisted at least for two days, suggesting prolonged circulation in the blood. This tumor accumulation of KR12-TAMRA was remarkable when most low-molecular-weight chemicals distribute all over various organs when evaluated in this assay. Furthermore, our results with nanoparticles showed distribution in the liver and kidney in addition to the tumor [19].

Tumor accumulation of PIP compounds has been reported [7,8]. Raskatov et al. [7] examined three different PIP compounds and found that one of these compounds exhibit preferential tumor accumulation. By analyzing ten-ring PIP compounds with a different structure, Inoue et al. suggest that the primary structure of PIP is important [8]. Examination of various PIP species for their tumor accumulation in the CAM model may provide insight into what contributes to the excellent tumor accumulation of KR12-TAMRA. We have noticed that KR12-TAMRA tumor accumulation continues for two days, suggesting its prolonged blood circulation. It is possible that KR12-TAMRA leaks out at the site of the tumor and accumulates in the tumor. Many macromolecules, such as the plasma proteins and lipids, tend to leak out from blood vessels in tumor tissue possibly due to the so-called enhanced permeation and retention (EPR) effect. KR12-TAMRA may escape blood circulation and tumor retention may be promoted by the binding of KR12-TAMRA to DNA in cancer cells. In fact, we have shown that KR12-TAMRA exhibits remarkable nuclear localization capability (Figure 6).

KR12 has been developed as an inhibitor of the *KRAS* gene expression [12]. Our study uncovered that the PIP contained in this inhibitor exhibits excellent tumor accumulation property. This PIP can be used in the future to derive a variety of KRAS inhibitors. Instead of CBI, a different alkylating agent or DNA damaging agent could be conjugated. Thus, our study forms the basis for future development of various KRAS inhibitors.

The CAM assay provides a promising method to evaluate the tumor accumulation property of various materials. We have previously shown that mesoporous silica-based nanoparticles exhibit tumor accumulation in the CAM model. In this paper, we showed that the tumor accumulation of PIP-based chemical can be evaluated by the CAM assay. Other types of materials can also be tested in the CAM assay. We as well as others have recently shown that CAM tumor can be formed using cancer patient-derived materials, such as surgical specimens of biopsy samples [20,22,24,25]. Thus, CAM models mimicking patient tumors can be established and these can be used to evaluate the tumor accumulation of various materials.

## 5. Conclusions

KR12 is a PIP-based inhibitor of *KRAS* gene expression. We synthesized KR12-PIP conjugated to TAMRA. Use of the CAM assay to evaluate the tumor accumulation of KR12-TAMRA uncovered the remarkable ability of this PIP to accumulate in the tumor. In addition, we observed the nuclear localization of this material.

## Figures and Tables

**Figure 1 cancers-14-00951-f001:**
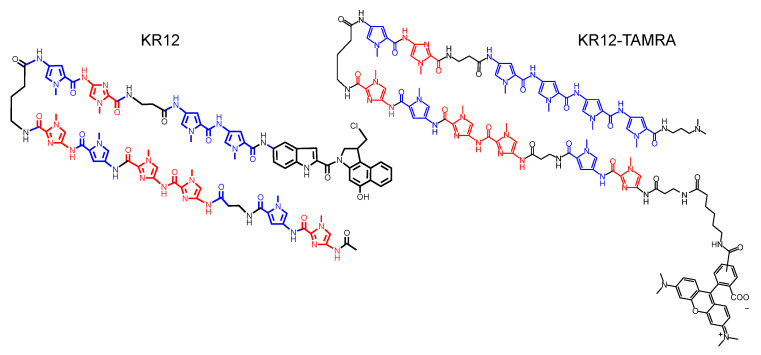
Chemical structure of KR12 and KR12-TAMRA.

**Figure 2 cancers-14-00951-f002:**
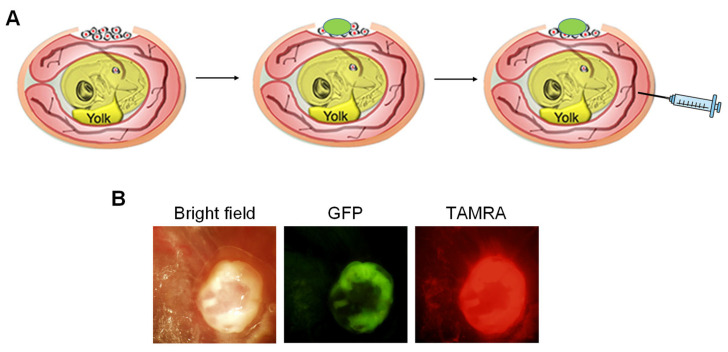
An approach to evaluate tumor accumulation of KR12-TAMRA. (**A**) An overview of our approach to examine tumor accumulation of KR12-TAMRA. (**B**) After intravenous injection of KR12-TAMRA into fertilized chicken egg transplanted with GFP expressing OVCAR8 cells, the green fluorescence of the tumor overlaps with the red fluorescence of KR12-TAMRA.

**Figure 3 cancers-14-00951-f003:**
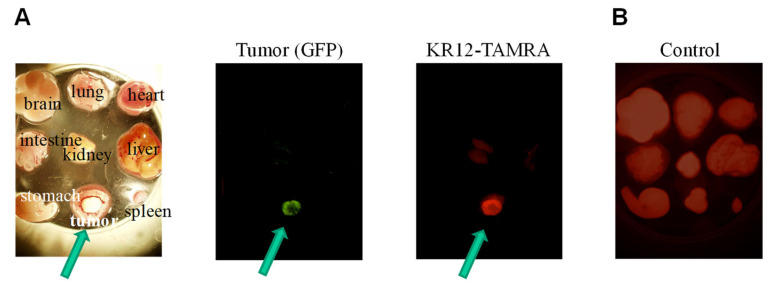
Preferential tumor accumulation of KR12-TAMRA. (**A**) Distribution of KR12-TAMRA was examined by its TAMRA fluorescence one day after the injection. (**B**) Control shows the red fluorescence of the RITC injected.

**Figure 4 cancers-14-00951-f004:**
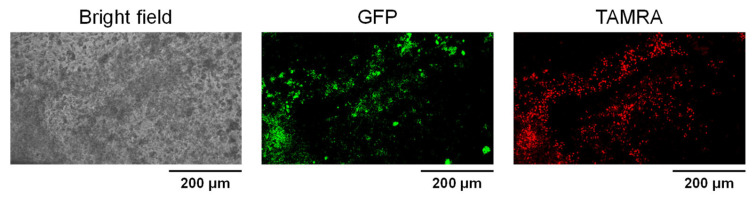
Overlap of KR12-TAMRA red fluorescence with the green fluorescence of cancer cells in the tumor.

**Figure 5 cancers-14-00951-f005:**
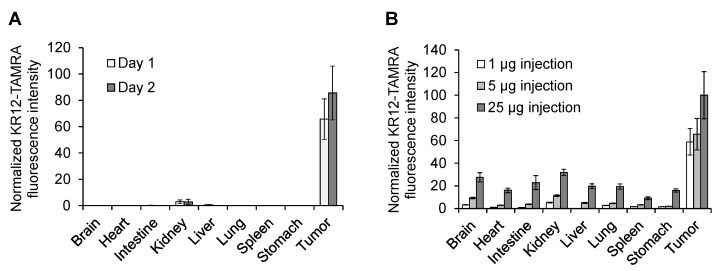
Tissue distribution of KR12-TAMRA on tumors and organs in fertilized chicken eggs transplanted with OVCAR8 cells. (**A**) Time course of tumor accumulation. Tissue distribution 1 or 2 days after injection of 5 µg of KR12-TAMRA. Mean ± SEM (*n* = 3–6). There were statistically significant differences between the tumor and each organ (*p* < 0.05). (**B**) Tissue distribution 1 day after injection of different concentrations of KR12-TAMRA. Mean ± SEM (*n* = 5–7). There were statistically significant differences between the tumor and each organ (*p* < 0.05).

**Figure 6 cancers-14-00951-f006:**
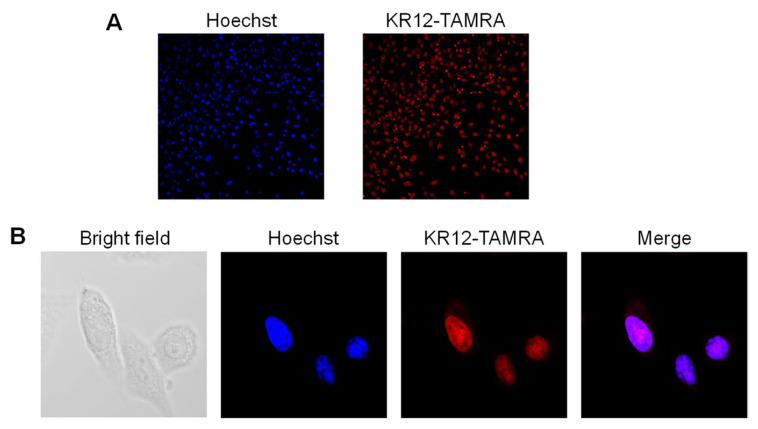
Nuclear localization of KR12-TAMRA. (**A**) Uptake of KR12-TAMRA into lung cancer cells examined at low magnification. (**B**) Higher magnification shows three cells in this field.

**Figure 7 cancers-14-00951-f007:**
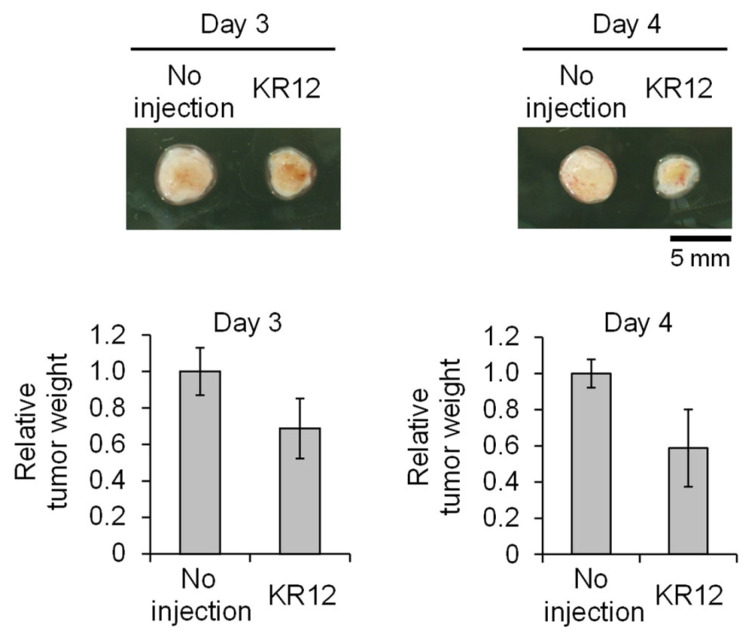
Tumor growth inhibition of KR12 in the CAM assay. Tumor photo as well as relative tumor weight are shown. A549 CAM tumors after injection of 5 µg of KR12. Values are presented as the mean ± SEM (*n* = 3).

## Data Availability

The data presented in this study are contained within the article or Appendix A.

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
