# Peer review of "Tumor Accumulation of PIP-Based KRAS Inhibitor KR12 Evaluated by the Use of a Simple, Versatile Chicken Egg Tumor Model"

_cancers, 2022, doi:10.3390/cancers14040951_

Round 1

Reviewer 1 Report

In this paper, the authors presented the synthesis of a derivative of KRAS inhibitor KR2 with a fluorescent label. The efficacy of tumor suppression of the synthesized compound was tested in a chicken egg model. From the biodistribution of the tumor model in a chicken egg, it was found that the compound was mainly localized in the tumors, which was very different from other small molecules, which distributed all over the organs. In the in vitro assay, it was found that the compound exhibited nuclear localization behavior. From the tumor assay, it was found that the growth of the tumor was suppressed by the injection of their compound. In, general, we found this paper very interesting. The results are very surprising that such small molecules exhibit specific accumulation in tumors, which is very important for the development of anticancer drugs. We have no problem in recommending publishing this paper. However, we have a few minor questions for the authors.   

  1. What is the mechanism for the specific accumulation of these molecules in the tumor?
  2. In figure 5 a, the injection amount of the drug is 5 ug on day 1 and day 2. Why the same amount of drugs shows slightly different biodistribution in figure 5b. In figure 5 a, almost nothing was observed on other organs except kidney. However, in figure 5b, we see a small percentage of drug accumulation in other organs.
  3. In figure 7, the relative tumor weight for day 3 and day 4 after treating with KR12 seems to be similar. Is there any statistical significance?
  4. Line 218, 219: figure 5a and figure 5b should be replaced by figure 6a and figure 6b.

Author Response

Response to reviewer 1

Comment 1: What is the mechanism for the specific accumulation of these molecules in the tumor?

Answer: We think that this has something to do with the so-called EPR (enhanced permeation and retention mechanism) effect. Because of breaks in tumor vasculature, many macromolecules such as plasma proteins and lipids tend to accumulate. KR12-TAMRA may escape blood circulation at tumor sites. Their possible association with macromolecules in the tumor may enhance tumor accumulation. We described this idea in the Discussion session in the revised manuscript.

Comment 2: In figure 5a, the injection amount of the drug is 5 μg on day 1 and day 2. Why the same amount of drugs shows slightly different biodistribution in figure 5b. In figure 5a, almost nothing was observed on other organs except kidney. However, in figure 5b, we see a small percentage of drug accumulation in other organs.

Answer: It seems to us that only a small percentage of drug accumulation is seen in other organs in both Figure 5A and 5B. The slight difference may be due to variability of very low- level signal which is near the detection limit. This is mentioned.

Comment 3: In figure 7, the relative tumor weight for day 3 and day 4 after treating with KR12 seems to be similar. Is there any statistical significance?

Answer: Thank you for your comment. In this experiment, we did not see much difference between day 3 and day4. However, we repeated this experiment to observe tumor growth inhibition by KR12 in the CAM assay. To avoid any confusion, we removed the word "significantly".  

Comment 4: Line 218, 219: figure 5a and figure 5b should be replaced by figure 6a and figure 6b.

Answer: Thank you for your comment. We corrected these in the revised manuscript.

Reviewer 2 Report

In this work Authors describe a chicken egg tumour model (CAM model) to grow xenografted cancer cell lines in order to examine tumour accumulation of biodegradable mesoporous silica nanoparticles. By showing the selective accumulation of the bound PIP portion of KR12 conjugated to fluorescent TAMRA into the tumour cell nuclei (but not in the chick embryo) and its effect on tumour cell proliferation after simple intravenous administration, they present an experimental design appropriate for demonstrating the specific accumulation in the tumour and the potential of CAM model for drug studies and nanocarriers.

This work presents some original aspects relevant in the field of drug delivery and new materials studies, although the lack of description of detailed statistical methods limits the consistency of the paper.

In my opinion, the paper deserves the publication in this Journal, but minor revisions are required to complete and support the reliable description of the data.  

Minor revisions:

- Statistical analysis description is missing. Please, describe in the Materials and Methods section all the tests that have been adopted, together with the numerosity of the samples and their relative significance.

- Please show in the legends the significant values of their results.

- From Line 61 to 65 there are too many technical details that are not needed in the Introduction section. Please highline only the introductive information at the base of their work

- at line 179 Please correct the continuity of the text

- At line 183, at the end of the caption, there are two dots

- At line 204 please replace @@ characters with appropriate units

- Instead of using “Biodistribution”, a pharmacokinetics constant with a very specific meaning, I would suggest to use the more appropriate term “tissue distribution”.

Author Response

Response to reviewer 2

Comment 1: Statistical analysis description is missing. Please, describe in the Materials and Methods section all the tests that have been adopted, together with the numerosity of the samples and their relative significance.

Answer: Thank you for your suggestions. We added a new section "2.7. Statistical Analysis" in the Materials and Methods in the revised manuscript where we described the details about the statistical analysis.

Comment 2: Please show in the legends the significant values of their results.

Answer: We described the significant value of statistical analysis in the legend.

Comment 3: From Line 61 to 65 there are too many technical details that are not needed in the Introduction section. Please highline only the introductive information at the base of their work.

Answer: Thank you for your suggestions. We rewrote this part to remove the technical details.

Comment 4: At line 179, please correct the continuity of the text.

Answer: Thank you for your comment. We corrected the continuity of the text in the revised manuscript.

Comment 5: At line 183, at the end of the caption, there are two dots.

Answer: Thank you for your comment. We removed one dot in the revised manuscript.

Comment 6: At line 204 please replace @@ characters with appropriate units.

Answer: Thank you for your comment. We corrected these characters to “μ” in the revised manuscript.

Comment 7: Instead of using “Biodistribution”, a pharmacokinetics constant with a very specific meaning, I would suggest to use the more appropriate term “tissue distribution”.

Answer: Thank you for your suggestion. We changed all "biodistribution" to "tissue distribution" in the revised manuscript.